# Study protocol for a randomized controlled trial comparing pulse pressure variation (PPV) and central venous pressure (CVP) guidance for fluid responsiveness assessment in neurosurgical patients undergoing posterior fossa tumor resection in park bench position

**Pathomporn Pin on** [ORCID]*, **Srisuluk Kacha, Ananchanok Saringkarinkul, Nakan Thanakititham**

Department of Anesthesiology, Faculty of Medicine, Chiang Mai University, Chiang Mai, Thailand

* pinon.pathomporn@gmail.com

## Abstract

### Introduction

Suboccipital craniotomy in the park bench position is linked to considerable physiological alterations. Effective fluid management in this context is a challenge to anesthesiologists. No published data exist on pulse pressure variation and central venous pressure guidance in patients undergoing tumor resection in the park bench posture. We undertake a study to evaluate the concept that two approaches of goal-directed fluid treatment enhance optimal fluid management and improve hemodynamic stability. We established the fluid management protocol for this process.

### Methods and analysis

This is a prospective randomized double-blinded study of adult patients undergoing suboccipital craniotomy to remove tumors in the park bench position. The comparison of pulse pressure variation and central venous pressure for fluid management regarding mean intraoperative fluid administration as the primary outcome. A sample size of 54 will yield over 80% power to identify a mean fluid difference of at least 500 ml between two specified methods. The secondary outcomes are data pertinent to fluid administered during and after surgery, including the lowest systolic blood pressure, serum lactate levels, vasopressor utilization, and duration of ICU stay. The statistical analysis will be validated based on the data distribution and types of data. This is the first study to examine two goal-directed fluid therapies, pulse pressure variation and central venous pressure, in patients with posterior fossa tumors and undergoing surgery in the park bench position. Researchers want to contribute novel information to the domain of fluid optimization in neurosurgery.

**Data availability statement:** Upon the completion of the study, the results will be presented in the full manuscript and publish in the journal.

**Funding:** The author(s) received no specific funding for this work.

**Competing interests:** The authors have declared that no competing interests exist.

## Trial registration

ClinicalTrials.gov NCT06595667.

## Introduction

Fluid management plays an important role in the perioperative care of neurosurgical patients, particularly those undergoing posterior fossa tumor resection [1,2]. In these procedures, patients are often placed in the park bench position [3,4]. The park bench position is a lateral positioning technique in which the head is oriented toward the surgical side, while the arm on the contralateral side of the lesion dangles off the operating table. The park bench position facilitates surgeons' access to pathology in the parietal, temporal region, and cerebellar-pontine angle (CPA) [5]. Surgery in the park bench position induces various physiological alterations, including oxygenation-ventilation matching, and hemodynamics [3,5]. Evaluating the sufficiency of fluid replacement during surgery is difficult while the patient is in the park bench position. The accumulation of blood in the dependent part, inferior vena cava (IVC) compression, positive intrathoracic pressure, and decreased peripheral vascular resistance affect the accuracy of fluid replacement evaluation [4,5]. Additionally, surgery in the posterior fossa, which houses the brainstem, leads to hemodynamic fluctuations and complicates the interpretation of fluid responsiveness and volume replacement [6,7].

Intraoperative fluid management is crucial to avert problems such as cerebral hypoperfusion or brain edema, which can significantly impact patient outcomes [1]. Multiple methods, static and dynamic indices, have been used to guide fluid administration intraoperatively [8]. Two methods that are frequently employed are central venous pressure (CVP) guidance and pulse pressure variation (PPV) guidance [9–11]. PPV, derived from arterial pressure waveform analysis, reflects the dynamic changes in stroke volume induced by mechanical ventilation, offering a real-time assessment of fluid responsiveness and has been proposed as a reliable indicator of fluid responsiveness in various surgical settings [9]. CVP is the most frequently used preload index [12,13]. The reliability of using CVP values to guide fluid administration should be applied to patients with normal cardiac function. Otherwise, a trend of increasing CVP values during fluid loading does not consistently suggest an increase in cardiac output [12]. Previous studies have revealed that CVP possesses a comparatively low predictive value for fluid responsiveness [13,14]. However, extreme CVP values, specifically below 6 mmHg or beyond 15 mmHg, can provide information about how to modify fluid delivery during surgery [15].

Previous studies indicated that using PPV guidance for fluid management during craniotomy was more reliable than using CVP guidance [16–18]. Nevertheless, such investigations were carried out on supine patients undergoing brain surgery. The reliability of PPV-guided fluid management in those studies was shown by increased hemodynamic stability, decreased incidence of hypotension, and decreased intraoperative fluid requirement, compared to the CVP group [16–18]. As of now, no

comparative analysis has been conducted on PPV and CVP concerning fluid management in the park bench position. We developed a prospective, randomized, controlled trial aimed at comparing the fluid management strategies guided by PPV and CVP in patients undergoing sub-occipital craniotomy (SOC) in the park bench position.

## Methods and analysis

The study aimed to investigate whether there are significant differences in fluid responsiveness when using PPV compared to CVP guidance during posterior fossa tumor resection in adult neurosurgical patients positioned in the park bench setup. This study is conducted in compliance with the SPIRIT (Standard Protocol Items: Recommendations for Interventional Trials) guideline.

### Ethical considerations

This study protocol has been approved by the Institutional Ethical Committee of the Faculty of Medicine, Chiang Mai University (CMU), with the protocol number ANE-2567-0391. All participants must provide consent to engage in the study once the research team thoroughly elucidates the study methodology. The results of this study will be published in a peer-review journal. Participants involved in the research may directly request the study results from the researchers upon publication. No additional blood work is required beyond the standard practice for patients undergoing SOC to remove tumors. Participants will be monitored in accordance with established standards of anesthesia care and treated in compliance with the Declaration of Helsinki. Confidential information will be securely stored throughout the duration of the study and thereafter. Participants and their relatives will be provided with a comprehensive overview of the study protocol during the informed consent process. Individuals possess the right to decline participation without experiencing any adverse consequences regarding their treatment and standard of care.

The risk assessment generally encompasses the subsequent factors: 1. Surgical and Anesthetic Risks: Participants in this study will undergo posterior fossa tumor resection in the park bench position, which carries inherent surgical and anesthetic risks, including potential vascular injury, nerve injury, brain edema, respiratory complications, and hemodynamic instability. The procedures utilized in this research do not present additional risks beyond those typically associated with such surgeries. 2. Specific Risks Related to Research: The primary research intervention entails continuous hemodynamic monitoring through the use of central venous pressure (CVP) and pulse pressure variation (PPV). Complications associated with central venous and arterial catheters may arise during insertion or usage. These include pneumothorax, cardiac arrhythmias, arterial injury, infectious complications, and thrombotic complications. Preventing complications requires strict adherence to aseptic techniques during catheter insertion, appropriate site selection, utilization of ultrasound guidance to mitigate mechanical complications, and consistent monitoring of the catheter site.

### Patient and public involvement

There will be no healthy volunteers in this study. The inclusion criteria are (1) adult patients, aged 20–65 years, (2) diagnosis of posterior fossa tumor, (3) scheduled for SOC to remove tumor in the park bench position, and (4) the patient is willing to participate in the study. The day before the procedure, the anesthesiologist will evaluate the patient in the ward, clarify the study protocol, and obtain informed consent. The patients will be notified that they may withdraw from the study at any time without affecting the standard treatment they are entitled to receive. Patients participating in the study will be treated following the standards of the Declaration of Helsinki. Patients with active cardiac illness, arrhythmias, sepsis, vasoactive drug dependence, chronic obstructive airway disease, pulmonary hypertension, and the American Society of Anesthesiologists (ASA) status equal to or greater than III will be excluded. The anticipated duration of participant involvement is one-week following admission to the neurosurgical intensive care unit (ICU) after posterior fossa tumor excision.

### Research site and anticipated schedule

This research is a prospective, single-center, double-blind randomized trial performed at a tertiary care university hospital in Northern Thailand. The institutional ethics committee approved the study protocol from September 26, 2024, until September 25, 2025.The research team intends to prospectively gather patient data from December 1, 2024, to May 31, 2025, in the brain and spinal cord operating room at Maharaj Nakorn Chiang Mai Hospital, as well as postoperative data from the neurosurgical intensive care unit. Data gathering will conclude by June 2025, with outcome analysis expected to be finalized by December 2025. Should the research team determine, within 45 days prior to the expiration of the ethical committee approval date, that recruitment of the planned patient cohort may be unfeasible, we will submit a request for a 6-month extension of the research approval.

### Confidentiality

Patient information, such as names and hospital numbers, will be retained by the co-investigator, while patients will be deidentified and assigned a unique research identification number instead. The publication will not disclose the patient's identity; however, if required for the patient's safety, identifying information may be revealed.

### Randomization and blinding

The schedule of enrolment, interventions, and assessments is shown in Fig 1. Patients will be randomly allocated to either CVP (group 1) or PPV (group 2)-guided fluid administration in a 1:1 ratio (Fig 2). Block randomization of block sizes 4, 6, and 8 was used to guarantee that the patient would be randomly and uniformly assigned to each group. A unique randomization code was generated from the website (sealedenvelope.com) by an independent research assistant [19]. The randomized code is sealed in an opaque envelope and will be opened when the patient arrives in the operating room.

The patients will be blinded to the group allocation as they are under general anesthesia. Surgeons will not be aware of the specifics of the fluid replacement throughout the procedure, even if they can see what the anesthesiologist is monitoring. The anesthesiologists responsible for the patient cannot be blinded as they have to provide fluids according to the study protocol. The statistician analyzing the outcomes will be unaware of the group to which the data pertains.

In the event of a critical occurrence during surgery, such as cardiac arrest or significant arrhythmias resulting in hemodynamic instability, the anesthesiologist will promptly notify the surgeon and disclose the patient allocation group.

### Preoperative preparation and informed consent process

The patients will be assessed by anesthesiology residents a day before the surgery. They will be advised about fasting, anesthesia, and surgical steps, the possibility of intubation postoperatively, and admission to neurosurgical ICU. The study protocol will be informed and requested for consent. Certain medications, such as antihypertensives, steroids, antibiotics, and anticonvulsants, will be maintained until the day of surgery. Sedative or anxiolytic medications are not administered routinely but will be evaluated on a case-by-case basis.

### Anesthesia protocol

At the operation theater, the patients will receive standard ASA monitoring including electrocardiogram (ECG), pulse oximetry ($SpO_2$), non-invasive blood pressure, temperature, end-tidal $CO_2$ ($etCO_2$), Bi-Spectral index (BIS), direct arterial pressure, and central venous pressure (CVP). Two large-bore peripheral venous lines will be established. All patients will have general anesthesia with endotracheal intubation. The target control infusion (TCI), Schnider model, will be used for the total intravenous anesthesia (TIVA) technique. The effect-site concentration of propofol will be started at 6–8 $\mu g.ml^{-1}$. Tracheal intubation will be facilitated with cis-atracurium 0.2 $mg.kg^{-1}$ and then maintenance infusion dose at 2 $\mu g.kg^{-1}.min^{-1}$ throughout the procedure. Cisatracurium is only used for intubation and not for infusion in certain cases where

| | | STUDY PERIOD | | January 1, 2025 to May 31, 2026 | | | | | |
|---|---|---|---|---|---|---|---|---|---|
| | Enrolment | Allocation | Post-allocation: during surgery | | | | | ICU NeuroSx |
| TIMEPOINT | $-t_1$ | $t_0$ | $t_1$ | $t_2$ | $t_3$ | $t_4$ | $t_5$ | $t_x$ |
| **ENROLMENT:** | | | | | | | | |
| **Eligibility screen** | ✓ | | | | | | | |
| **Informed consent** | ✓ | | | | | | | |
| **Allocation** | | ✓ | | | | | | |
| | | | | | | | | |
| INTERVENTIONS: | | | | | | | | |
| *[Intervention A: PPV]* | | | ✓ | ✓ | ✓ | ✓ | ✓ | |
| *[Intervention B: CVP]* | | | ✓ | ✓ | ✓ | ✓ | ✓ | |
| | | | | | | | | |
| **ASSESSMENTS:** | | | | | | | | |
| *[Baseline variables]* | ✓ | ✓ | | | | | | |
| *[ Primary outcome]* | | | | ✓ | ✓ | ✓ | ✓ | |
| *[Secondary outcomes]* | | | | ✓ | ✓ | ✓ | ✓ | ✓ |

**Fig 1. The schedule of enrolment, interventions, and assessments.**

surgeons need to monitor cranial nerves, such as tumor resection in the CPA region, since it could interfere with the assessment of motor function of those cranial nerves. A mixture of oxygen and air will be used during anesthetization. The patients will receive an intermittent bolus dose of fentanyl, 2–3 µg.kg$^{-1}$. Post-induction, a 20-G arterial catheter will be placed into the radial artery of the arm corresponding to the side of the lesion, which will be positioned on top when the patient is arranged in the park bench position. A 7-Fr, triple-lumen central venous catheter will be inserted in the right subclavian vein, under ultrasound guidance, for the CVP monitoring. Plain bupivacaine 0.5%, 10 ml, will be injected over the nerves supplying the scalp on the surgical side, including the pre-auricular nerve and the greater and lesser occipital nerves. The anesthesiologist will provide 2 ml of 2% lidocaine at each of the three pre-marked locations, five minutes before the surgeon applies the Mayfield head holder. The neuro-surgical resident will insert a urinary catheter. The patient will be carefully positioned on the park bench. The endotracheal tube, intravenous line, arterial line, and CVP line will be re-checked after the position is finalized. The ventilator setting (Maquet Critical Care AB, Sweden) is tidal volume 8 ml.kg$^{-1}$(ideal body weight), respiratory rate will be adjusted to keep normocarbia (etCO$_2$ 35–40 mmHg), inspiration: expiration ratio 1:2, positive end-expiratory pressure (PEEP) 5 cmH$_2$O, and a fraction of inspired oxygen of 0.5 with air mixture. The intraoperative parameters including heart rate, arterial blood pressure, PPV, and CVP, will be recorded on the Mindray, BeneVision N12 (Shenzhen Mindray Bio-Medical Electronics). The anesthetic depth will be monitored using a BIS monitor (Covidien Ireland Limited) and maintained between 40–60 throughout the operation. Serum blood lactate levels will be evaluated hourly using arterial blood gas analysis throughout the surgical procedure. Mannitol is not routinely

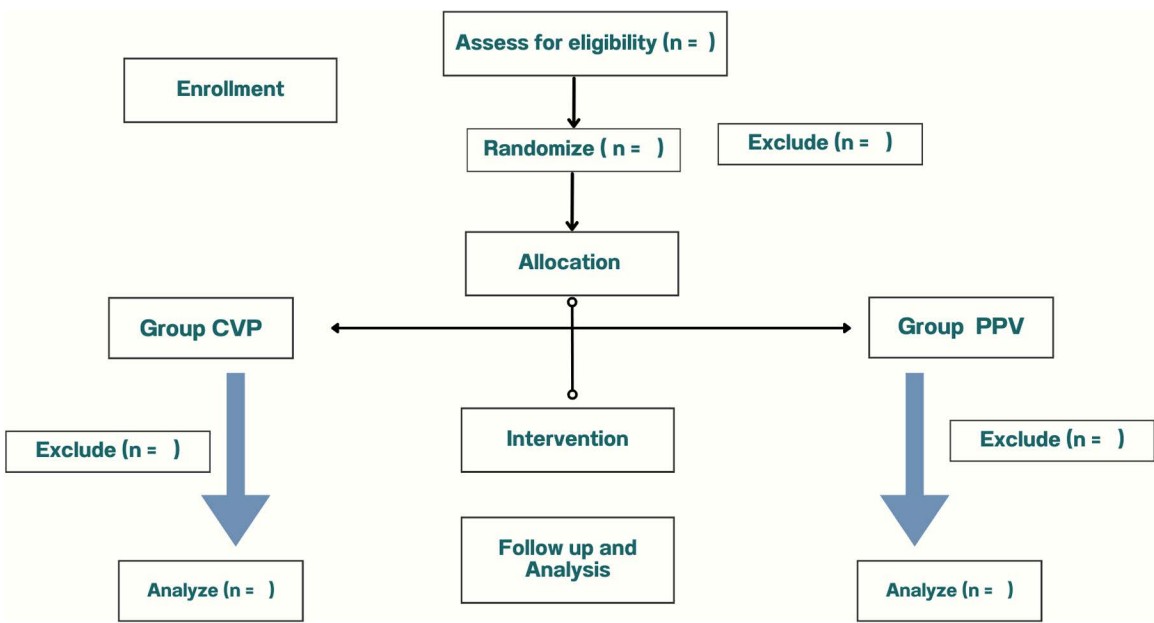

**Fig 2. The Consolidated Standards of Reporting Trials flow diagram.**

given to every patient; it depends on the surgeon's decision. During the surgery, the patient will receive fluids following the fluid management protocol, contingent upon their classification into the PPV or CVP fluid guidance group. At the end of the surgery, should the patient receive muscle relaxants, they will be antagonized with neostigmine at 0.05 mg. kg-1 and glycopyrrolate at 0.01 mg.kg⁻¹. The patient must meet the safe extubation criteria; otherwise, if the surgeon determines that extubation is premature, the patient will stay intubated state and be transferred to the neuro-ICU. The surgeon will be asked to rate the brain relaxation, using a four-point scale (1 = relax, 2 = satisfy, 3 = firm, 4 = bulging) [20]. At the ICU, the anesthesiologist will examine the conjunctival edema, peri-orbital swelling, and the cuff leakage test, if the patient is in the intubated state.

## Fluid management protocol

CVP and PPV will be measured and recorded in both groups. Both parameters appear on the monitor screen throughout the operation. As a maintenance fluid, each patient will receive a weight-based formula of 0.9% normal saline solution and lactated Ringer solution [21]. Fluid management protocol will be activated, based on their assigned group (CVP or PPV), as shown in Fig 3. The treatment goal for the PPV group is a PPV value of less than 13%, whereas the treatment goal for the CVP group is a CVP value between 8–12 mmHg [22,23]. Before the implementation of the fluid management protocol, acute blood loss exceeding the permissible threshold and/ or hemoglobin < 9 gm/dl must be addressed and treated with pack red cell (PRC) transfusion [24]. The maximum allowable blood loss (MABL) is determined by considering gender, weight-based estimated blood volume, preoperative hematocrit, and the minimum acceptable hematocrit for the patient.

## Postoperative care

Post-surgery, all patients will be transferred to the neurosurgical ICU. The researchers will observe the patients for one week and will document data including the duration of intubation, the time point at which successful extubation, daily fluid balance, serum blood lactate, chest X-ray, and BUN/Cr values.

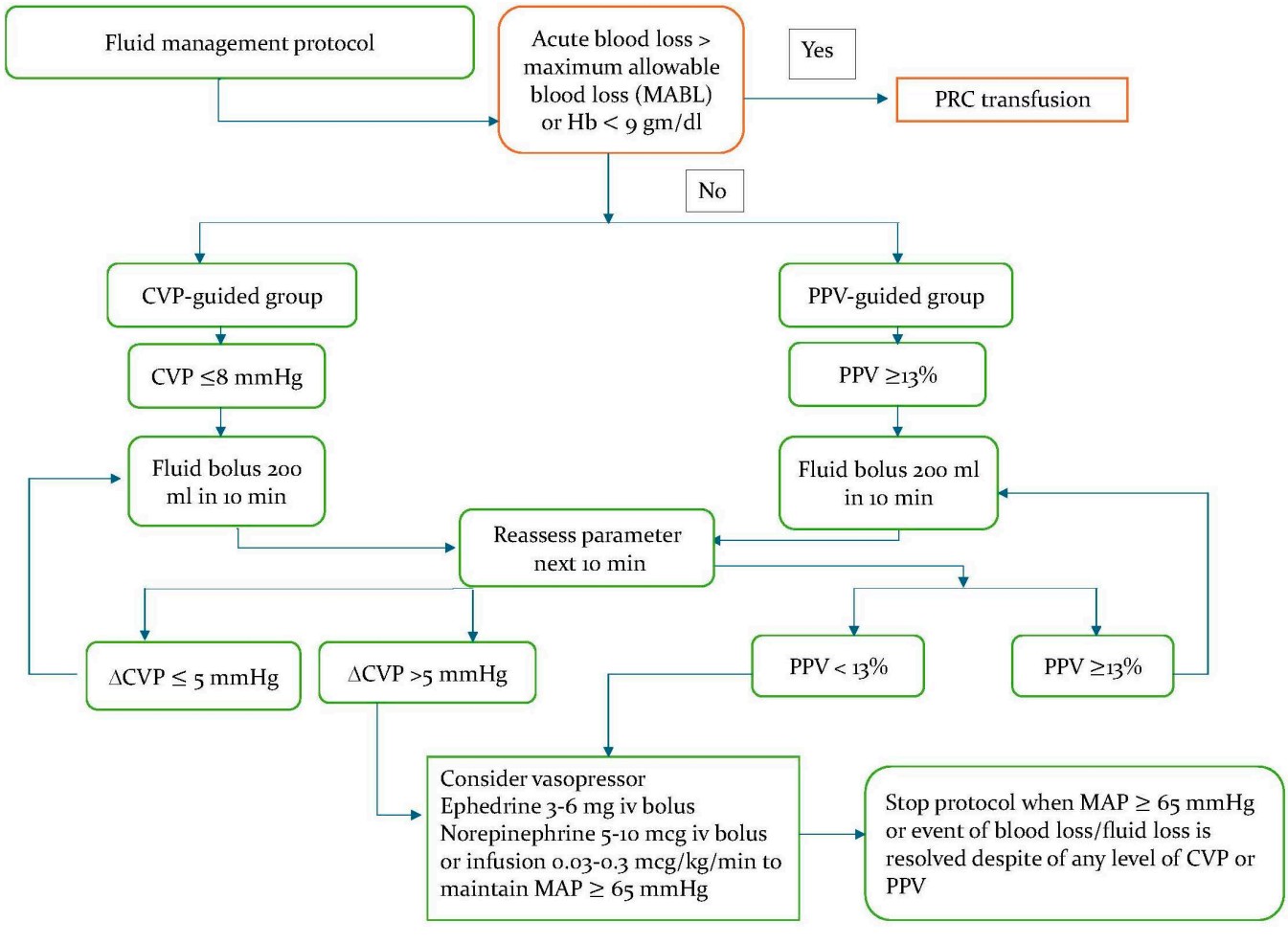

**Fig 3. Fluid management protocol.**

## Outcomes

**Primary outcome.** The primary outcome is the mean volume of intravenous fluids administered intraoperatively in the PPV group vs in the CVP group. The data will be represented in mean difference, 95% confidence interval, and p-value.

**Secondary outcomes.** Secondary outcomes are divided into intraoperative and postoperative data. Intraoperative data are (1) lowest systolic BP (mmHg), (2) blood loss during surgery (ml), (3) the requirement of vasopressors (no. and %), (4) urine output (ml), (5) serum lactate level (mg/dl), (6) brain relaxation score (percentage of a four-point scale), and (7) the incidence of extubation in the operating theater (no. and %). Postoperative data will be collected at the neurosurgical ICU, including (1) the length of time the patient is mechanical ventilation dependent (hours), (2) serum blood lactate, (3) chest X-ray, and BUN/Cr values and (4) The duration of the patient's stay in the ICU (day).

Secondary outcomes are categorized as either confirmatory or exploratory to guide interpretation. Confirmatory secondary outcomes include serum lactate levels (intraoperative and postoperative), duration of ICU stay, and lowest systolic blood pressure. These outcomes will be analyzed with adjustments for multiple comparisons using Bonferroni correction. The remaining secondary outcomes, comprising less critical intraoperative variables such as brain relaxation score or urine output and postoperative variables such as chest X-ray abnormalities, are designated as exploratory. Exploratory

outcomes will be reported descriptively and analyzed for hypothesis generation, without corrections for multiple comparisons.

## Data collections and data management

The non-endpoint data include patient characteristics and details regarding the operation and anesthesia. Types and dosages of anesthetic and opioids administered during surgical procedures, hemodynamic data, mannitol administration, use of inotropic or vasopressor agents, volume of blood transfusion, estimated blood loss, urine output, and duration of surgery will be recorded in the case record form (CRF). Two independent anesthesiologists, blinded to the group allocation, will check the data, patient safety, the accuracy and safety of data, and the study progression once a month.

## Sample size calculation

The primary outcome is the intraoperative fluid volume administered. The calculation of sample size utilized data from the research conducted by Janani Gopal et al [18]. The average fluid volume administration of patients in the CVP and PPV groups were 4,300 and 3,500 ml, respectively. In another study, the groups that used CVP guidance and those that used PPV guidance had a mean intraoperative fluid volume difference of 556 milliliters [25]. With the power of 80%, a 0.05 two-sided difference, the calculated sample size was 25 patients/group. Given a dropout rate of 5%, we will enroll 54 patients. The website clincalc.com is used to calculate the sample size. We anticipate a minimal drop-out and low loss to follow-up rate, as the study is conducted while the patient is under general anesthetic during surgery. Additionally, post-surgery, all patients will be admitted to the neurosurgical ICU, where we can gather data regarding the duration of their ICU stay.

## Statistical analysis

Baseline variables will be summarized using descriptive statistics (mean, standard deviation, median, interquartile range, or frequency and percentage, as appropriate). A modified intention-to-treat (ITT) principle will be used for all analyses. Data distribution will be assessed using the Kolmogorov-Smirnov test. Comparisons between groups (PPV vs. CVP) will be conducted as follows:

**Primary outcome.** The primary outcome, the volume of intravenous fluid administered at multiple time points (t2 to t5), will be analyzed using a linear mixed-effects model (LMM) to account for repeated measures. Group (PPV vs. CVP), time (t2–t5), and interaction between group and time will be included as fixed effects, with a random intercept to account for within-patient correlation. Covariates will also be incorporated as fixed effects to adjust for potential baseline imbalances. For non-normal data, the generalized estimating equations (GEE) will be used.

**Secondary outcomes.** *Intraoperative data:* Continuous variables (e.g., lowest systolic BP, blood loss, urine output) will be analyzed using an independent sample t-test or Mann-Whitney U test, as appropriate. Covariates will be adjusted using an analysis of covariance (ANCOVA). Categorical variables (e.g., vasopressor use, incidence of extubation) will be compared using the Chi-square test or Fisher's exact test. For ordinal variables (e.g., brain relaxation score), a Mann-Whitney U test will be used.

***Postoperative data:*** For time-dependent outcomes (e.g., duration of mechanical ventilation, ICU length of stay), a Cox proportional hazards model will be used to estimate hazard ratios between groups (PPV vs. CVP), including baseline covariates to control for potential confounding. Kaplan-Meier survival analysis will be performed for graphical representation. Continuous variables (e.g., serum lactate levels, BUN/Cr values) will be compared using a t-test or Mann-Whitney U test, while categorical variables (e.g., chest X-ray abnormalities) will be compared using Chi-square or Fisher's exact test. For repeated measures within the ICU (e.g., serum lactate levels over time), a linear mixed-effects model (LMM) will be applied.

## Adjustments and multiple comparisons

To adjust for potential baseline covariate imbalances, specific analyses including ANCOVA, Cox proportional-hazards model, or LMM) will be used. To account for multiple comparisons across all secondary endpoints, a Bonferroni correction adjustment, will be applied.

A two-sided p-value of <0.05 will be considered statistically significant for the primary endpoint. The exploratory secondary analyses will report unadjusted p-values to guide future research. All statistical analyses will be performed using SPSS software by an expert statistician.

## Discussion

Optimizing fluid administration during neurosurgery poses a challenge to anesthesiologists. Fluid overload leads to brain edema [26]. Constrained cerebral volume within limited cranial capacity complicates adequate tumor resection for surgeons and may result in further injury from retraction pressure [3–8]. While inadequate fluid supply leads to hemodynamic instability and compromised tissue perfusion and oxygen delivery. Anesthesiologists must consider multiple factors, including osmotic diuresis, administration of mannitol, preoperative dehydration due to impaired consciousness, intraoperative blood loss, as well as surveillance for brain edema [17,27]. Several studies have previously compared fluid administration in brain surgery using PPV and CVP guidance; however, these studies focused on patients undergoing surgery in the supine and lateral positions [16–18]. Sandaram et al. concluded that PPV is a reliable indicator for fluid management in neurosurgical patients undergoing tumor removal in both supine and lateral positions [16]. A further study supporting the utilization of PPV fluid guidance in neurosurgical patients undergoing supratentorial tumor surgery in the supine position is the research conducted by Gopal et al [18]. It has been demonstrated that patients who received fluids guided by PPV had better hemodynamic stability, more satisfied brain conditions, and no compromise of perfusion compared to patients who received fluids guided by CVP [18]. Bhokare et al. conducted a diagnostic accuracy assessment, concluding that PPV fluid guiding is superior to CVP, which results in excessive fluid delivery in neurosurgical patients [17]. No study has previously examined the application of PPV or CVP fluid guidance in surgeries conducted in the park bench position. In conclusion, this study is the first to offer additional support for goal-directed fluid therapy, PPV vs. CVP guidance, in patients undergoing posterior fossa brain tumor surgery while employing the park bench position.

## Supporting information

**S1 File. The original study protocol version 2.0 submitted to the institutional ethical committee.**
(DOCX)

**S2 File. The completed SPIRIT checklist.**
(DOCX)

**S3 File. The human participants research checklist.**
(DOCX)

## Author contributions

**Conceptualization:** Pathomporn Pin on.

**Investigation:** Pathomporn Pin on, Srisuluk Kacha, Ananchanok Saringkarinkul, Nakan Thanakititham.

**Methodology:** Pathomporn Pin on, Srisuluk Kacha, Ananchanok Saringkarinkul, Nakan Thanakititham.

**Writing – original draft:** Pathomporn Pin on.

**Writing – review & editing:** Pathomporn Pin on, Srisuluk Kacha.

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
