## [Decision Letter · Decision Letter 0]

16 Mar 2025

PONE-D-24-50639Study protocol for comparing pulse pressure variation and central venous pressure guidance for fluid responsiveness assessment in neurosurgical patients undergoing posterior fossa tumor resection in park bench positionPLOS ONE

Dear Dr. Pin on,

Thank you for submitting your manuscript to PLOS ONE. After careful consideration, we feel that it has merit but does not fully meet PLOS ONE’s publication criteria as it currently stands. Therefore, we invite you to submit a revised version of the manuscript that addresses the points raised during the review process.

We look forward to receiving your revised manuscript.

Kind regards,

Imtiaz Wani

Academic Editor

PLOS ONE

Journal Requirements:

2. We note that you have indicated that there are restrictions to data sharing for this study. For studies involving human research participant data or other sensitive data, we encourage authors to share de-identified or anonymized data. However, when data cannot be publicly shared for ethical reasons, we allow authors to make their data sets available upon request. For information on unacceptable data access restrictions, please see http://journals.plos.org/plosone/s/data-availability#loc-unacceptable-data-access-restrictions .   

b) If there are no restrictions, please upload the minimal anonymized data set necessary to replicate your study findings to a stable, public repository and provide us with the relevant URLs, DOIs, or accession numbers. Please see http://www.bmj.com/content/340/bmj.c181.long for guidelines on how to de-identify and prepare clinical data for publication. For a list of recommended repositories, please see https://journals.plos.org/plosone/s/recommended-repositories. You also have the option of uploading the data as Supporting Information files, but we would recommend depositing data directly to a data repository if possible

3. We note that the original protocol that you have uploaded as a Supporting Information file contains an institutional logo. As this logo is likely copyrighted, we ask that you please remove it from this file and upload an updated version upon resubmission.

4. Please include captions for your Supporting Information files at the end of your manuscript, and update any in-text citations to match accordingly. Please see our Supporting Information guidelines for more information: http://journals.plos.org/plosone/s/supporting-information .

Additional Editor Comments :

Statistical analysis needs to be more refined

Reviewers' comments:

Reviewer's Responses to Questions

**Comments to the Author**

1. Does the manuscript provide a valid rationale for the proposed study, with clearly identified and justified research questions?

Reviewer #1: Yes

Reviewer #2: Yes

2. Is the protocol technically sound and planned in a manner that will lead to a meaningful outcome and allow testing the stated hypotheses?

Reviewer #1: No

Reviewer #2: Yes

3. Is the methodology feasible and described in sufficient detail to allow the work to be replicable?

Reviewer #1: No

Reviewer #2: Yes

4. Have the authors described where all data underlying the findings will be made available when the study is complete?

Reviewer #1: Yes

Reviewer #2: Yes

5. Is the manuscript presented in an intelligible fashion and written in standard English?

Reviewer #1: Yes

Reviewer #2: Yes

6. Review Comments to the Author

You may also provide optional suggestions and comments to authors that they might find helpful in planning their study.

Reviewer #1: This is a comparative study utilizing an ITT approach with a sample size requirement of 54 for the primary endpoint. The power to compare the two interventions will be about 0.80 for a two sided alpha of 0.05. In general, the statistical design and analysis have been well thought out. There are some issues.

On lines 38 and 39 of this document the authors state that,’ This study is methodologically sound, with a sample size determined to possess adequate power to identify variations in the research inquiries. A representative sample size reinforces external validity.’ Upon further examination of the statistical design, one sees that the sample size will suffice for the primary endpoint. However, the secondary outcomes are divided into intraoperative and postoperative data. There are over 7 intraoperative endpoints and 4 postoperative inquiries as well. Depending on the priority of these secondary endpoints, and without adjustment of the p-values in the analyses, that is multiple comparisons, then the type one error of 0.05 will be compromised for the secondary outcomes. This should be addressed in the statistical analysis section. Also, what does the statement,’ A representative sample size reinforces external validity.’ mean?

Also, in the Methods section and on Figure 2 one sees assessments at times t2 to t5. Back in the Statistical analysis section (lines 215 to 228) the investigators state that ,’ The primary outcome, the volume of intravenous fluid between the two groups (CVP vs. PPV), will be analyzed using an independent sample t-test, or the Mann-Whitney U test if the data is not normally distributed.’ It appears that this may be an oversimplification of the analysis. One appears to have repeated measures (assessment times)requiring a more involved multi way ANOVA or non parametric equivalent or more general linear model. That is, a group by time (with possible interaction) situation. The two sample test may be inadequate. Please rethink this.

Also in this same statistical section the authors state that ,’ We will use a regression approach to adjust the effect of covariate imbalance.’ Does one mean an analysis of covariance? Please be specific. Alternatively, the time-dependent variables for the fluid management protocol will use the COX model on these outcomes, as the authors note, which should suffice.

Reviewer #2: the article titled Study protocol for comparing pulse pressure

variation and central venous pressure guidance for fluid responsiveness assessment in neurosurgical patients

undergoing posterior fossa tumor resection in park bench position is innovative and submitted explanation to majority of the suggestions. it also forms a baseline for future research.

7. PLOS authors have the option to publish the peer review history of their article (what does this mean? ). If published, this will include your full peer review and any attached files.

**Do you want your identity to be public for this peer review?** For information about this choice, including consent withdrawal, please see our Privacy Policy .

Reviewer #1: No

Reviewer #2: No

---

## [Author Response · Author response to Decision Letter 1]

14 Apr 2025

Reviewer #1 raised the concern about the multiple testing issues and the potential inflation of the type I error rate (false-positive rate) for the secondary endpoints. This manuscript has many secondary endpoints which divided into intraoperative (over 7 endpoints) and postoperative (4 endpoints) inquiries.

To respond to this concern:

I have revised the manuscript, in the statistical analysis section, by addressing the inflation of type I error due to multiple comparisons across secondary endpoints by using the Bonferroni correction. With this adjustment method, I have to balance the type I error control while maintaining statistical power across diverse endpoints.

Additionally, I have prioritized some of the secondary outcomes as “highly clinical importance” (confirmatory) to merit statistical control. For the rest of them, I have labelled as “hypothesis-generating” (explanatory) secondary endpoints in order to reduce expectations of rigorous type I error control.

In the limitation section, I have acknowledged that “Adjustments for multiple comparisons will be applied for confirmatory analyses of secondary endpoints to control the type I error rate at 5% level. However, the explanatory secondary endpoints are acknowledged to carry risk of inflated type I error”.

For the controversy of this sentence ‘A representative sample size reinforces external validity.’. I decided to cut this sentence out. Instead, I added the clarification of statistical analysis as mentioned above.

We have updated the statistical analysis section to specify the use of Bonferroni correction to adjust for multiple comparisons across the secondary endpoints.

We have clarified the prioritization of secondary outcomes into confirmatory (with adjusted p-values) and exploratory analyses. This ensures transparency of our approach while maintaining scientific integrity.

Efforts to balance overall type I error control and interpretability of multiple endpoints have been discussed in the limitations section.

The reviewer has provided a valid and insightful critique regarding the analysis of primary outcome which will be assessed over multiple time points (t2-t5). Here is my revision.

“For the primary outcome, the volume of intravenous fluid between the two groups (CVP vs. PPV) at multiple time points (t2 to t5), we will use a repeated measures approach to account for intra-subject correlation over time. Depending on the data distribution:

For normally distributed data, a linear mixed-effects model (LMM) will be employed, including group (CVP vs. PPV), time (t2 to t5) and their interaction as fixed effects, and a random intercept for each subject to account for within-patient variability.

For non-normally distributed data, we will employ non-parametric alternative, generalized estimating equations (GEE). Interaction effects between time and group will also be evaluated to investigate differential effects over time.

The reviewer is asking for more specificity about the type of regression approach we plan to use for adjusting covariates. The phrasing "regression approach" is too broad. We have explicitly stated the exact method that will use based on the nature of the primary and secondary outcomes. For the primary outcome, which is continuous in nature and are measured across multiple time points, a linear mixed-effects model (LMM) will be accounted for both covariates and repeated measures.

"For other continuous outcomes such as blood loss, or serum lactate, where necessary, an analysis of covariance (ANCOVA) will be used to adjust for baseline covariates to examine group differences.

"For outcomes with time-to-event data, such as the duration of mechanical ventilation dependence or ICU stay, a Cox proportional-hazards model will be used to estimate the hazard ratio between groups (PPV vs. CVP). Baseline covariates, if imbalanced, will be included in the model to control for confounding effects, ensuring accurate estimation of treatment effects."We clarified that for continuous outcomes requiring adjustment for covariate imbalance, analysis of covariance (ANCOVA) will be used. For repeated measures, we specified the use of a linear mixed-effects model (LMM), which accommodates covariates as fixed effects while accounting for within-subject correlations.

• For time-dependent outcomes such as ICU length of stay and duration of mechanical ventilation, we confirmed the use of a Cox proportional-hazards model, which includes baseline covariates to control for potential confounding.

These updates ensure that our analysis plan is sufficiently rigorous and transparent. Thank you again for your invaluable feedback, which has greatly improved the robustness of our methodology.

The logo on the EC approval document has been removed.

The statistical analysis is major adjusted according to the reviewers' comments.

---

## [Decision Letter · Decision Letter 1]

29 Apr 2025

Study protocol for comparing pulse pressure variation and central venous pressure guidance for fluid responsiveness assessment in neurosurgical patients undergoing posterior fossa tumor resection in park bench position

PONE-D-24-50639R1

Dear Dr. Pin on,

We’re pleased to inform you that your manuscript has been judged scientifically suitable for publication and will be formally accepted for publication once it meets all outstanding technical requirements.

Kind regards,

Imtiaz Wani

Academic Editor

PLOS ONE

Additional Editor Comments (optional):

Revised manuscript deemspublishing

Reviewers' comments:

Reviewer's Responses to Questions

**Comments to the Author**

1. Does the manuscript provide a valid rationale for the proposed study, with clearly identified and justified research questions?

Reviewer #1: Yes

2. Is the protocol technically sound and planned in a manner that will lead to a meaningful outcome and allow testing the stated hypotheses?

Reviewer #1: Yes

3. Is the methodology feasible and described in sufficient detail to allow the work to be replicable?

Reviewer #1: Yes

4. Have the authors described where all data underlying the findings will be made available when the study is complete?

Reviewer #1: Yes

5. Is the manuscript presented in an intelligible fashion and written in standard English?

Reviewer #1: Yes

6. Review Comments to the Author

You may also provide optional suggestions and comments to authors that they might find helpful in planning their study.

Reviewer #1: Appropriate revisions have been made and additions to the protocol included.

XXXXXXXXXXXXXXXXXXXXXXXXXXXXXXXXXXXXXXXXXXXXXXXXXXXXXXXXXXXXXXXXXXXXXXXXXXXXXXXXXXXXXXXXXXXXXXXXXXXXXXXXXXXXXXXXXXXXXXXXXXXXXXXXXXXXXXXXXX

7. PLOS authors have the option to publish the peer review history of their article (what does this mean? ). If published, this will include your full peer review and any attached files.

**Do you want your identity to be public for this peer review?** For information about this choice, including consent withdrawal, please see our Privacy Policy .

Reviewer #1: No

---

## [Editor Report · Acceptance letter]

PONE-D-24-50639R1

PLOS ONE

Dear Dr. Pin on,

I'm pleased to inform you that your manuscript has been deemed suitable for publication in PLOS ONE. Congratulations! Your manuscript is now being handed over to our production team.

Kind regards,

on behalf of

Dr. Imtiaz Wani

Academic Editor

PLOS ONE